# Left atrial emptying fraction determined during atrial fibrillation predicts maintenance of sinus rhythm after direct current cardioversion in patients with persistent atrial fibrillation

Paweł Wałek[1], Iwona Gorczyca[1,2]*, Janusz Sielski[2,3], Beata Wożakowska-Kapłon[1,2]

1 1st Clinic of Cardiology and Electrotherapy, Voivodship Hospital Kielce, Kielce, Poland, 2 Collegium Medicum, Jan Kochanowski University, Kielce, Poland, 3 Intensive Cardiac Care Unit, Voivodship Hospital Kielce, Kielce, Poland

* iwona.gorczyca@interia.pl

**Data Availability Statement:** All relevant data are within the paper and its Supporting Information files.

## Abstract

Echocardiography is the basic imaging technique used to determine the odds of maintaining sinus rhythm (SR) following direct current cardioversion (DCCV) for persistent atrial fibrillation (AF). However, most studies are focused on the echocardiographic parameters obtained during SR resulting from successful DCCV. The aim of this study was to assess the value of the echocardiographic parameters measured before DCCV for the prognosis of SR maintenance after DCCV. The study included 146 patients with persistent AF who underwent DCCV. Clinical and echocardiographic data were collected directly before DCCV and, for patients with SR, one month, six months, and 12 months after DCCV. We found that left atrial emptying fraction (LAEF) assessed during atrial fibrillation was significantly larger in the group with SR maintenance after 12 months than in the group with AF recurrence (30.8±8.3 vs. 24.6±10.4%; p<0.001). In multivariable logistic regression analysis with a model containing echocardiographic parameters, LAEF (OR 1.053; 95% CI 1.011–1.096; p = 0.013) and the E/e'$_{mean}$ ratio (OR 0.883; 95% CI 0.788–0.990; p = 0.033) were independent predictors of SR maintenance. Analyzing a model including clinical and echocardiographic variables, only LAEF (OR 1.046; 95% CI 1–1.095; p = 0.049) and beta-blockers used before DCCV (OR 14.694; 95% CI 1.622–133.139; p = 0.017) were independent predictors of SR maintenance after 12 months. Our results indicate that LAEF measured during AF is a significant predictor of SR maintenance in the 12 months following DCCV due to persistent AF. Our findings confirm the recently raised hypothesis about the superiority of echocardiographic parameters assessing mechanical remodeling over parameters assessing structural remodeling of left atrium in predicting sinus rhythm maintenance after electrical cardioversion.

**Funding:** This project was funded by the Ministry of Science and Higher Education, under the program "Regional Initiative of Excellence" (Project no. 024/RID/2018/19; amount granted: 11,999,000 PLN). The funders had no role in study design, data collection and analysis, decision to publish, or preparation of the manuscript.

**Competing interests:** The authors have declared that no competing interests exist.

## Introduction

Atrial fibrillation (AF) is one of the most frequently diagnosed persistent supraventricular tachycardia [1] and one of the most common risk factors for cardiovascular diseases [2]. AF induces electrical, mechanical, and structural remodeling of the left atrial (LA) and right atrial (RA) myocardium [3]. Direct current cardioversion (DCCV) is one of the basic procedures applied to restore sinus rhythm (SR) and reduce symptoms in patients with AF and could be abridging therapy until AF ablation is performed in a selected group of patients [2]. As a result of DCCV, SR is restored in about 90% of cases, but it is maintained only in about 70% of cases after 12 months [4]. Clinical, echocardiographic, and biochemical parameters are currently sought to help assess the prognosis of SR maintenance after DCCV [5–7]. The echocardiographic parameters most frequently mentioned as prognostic factors for SR maintenance are those assessing the structural remodeling of the left atrium, including the LA antero-posterior diameter (LAAP) and the LA volume index (LAVI); those assessing mechanical remodeling, such as the LA emptying fraction (LAEF) assessed during SR; and those assessing the left ventricular filling pressure (LVFP) [8–19]. Most studies have focused on the echocardiographic parameters measured during SR, following a successful DCCV. In this study, we measured the echocardiographic parameters before DCCV, during AF, and analyzed their potential in predicting SR maintenance.

## Methods

### Study population

One hundred and forty-six patients with persistent AF who underwent DCCV in our Cardiology Division between August 2015 and April 2017 were prospectively enrolled in the study. Inclusion criteria were as follows: symptomatic persistent AF lasting a minimum of seven days; ejection fraction of more than 40%; and appropriate anticoagulation a minimum of three weeks before DCCV with warfarin, acenocoumarol, dabigatran, rivaroxaban, or apixaban. Exclusion criteria were as follows: age under 18 years, lack of consent for study participation, lack of consent for DCCV, poor quality of echocardiography visualization, ventricular rate greater than 120 beats per minute, moderate or severe valve regurgitation or stenosis, valvular prosthesis, the presence of thrombus in the left atrial appendage, acute decompensation of heart failure, acute myocardial infarction, previous pulmonary vein isolation, dysthyroidism, anemia with hemoglobin <6.9 mmol/l, and the presence of neoplastic disease. Clinical and echocardiographic data were collected directly before DCCV. Follow-up electrocardiograms and clinical data were collected from all patients with SR after one month, six months, and 12 months. A 24-hour ambulatory electrocardiographic monitoring was performed on all patients who were in SR at the one- and 12-month follow-ups. Patients were instructed to report to our Cardiology Department if they felt palpitations or had arrhythmia recurrence. The study protocol was approved by the Institutional Review Board of the Świętokrzyskie Medical Chamber.

### Clinical data

Clinical data were obtained on the day of DCCV and included the following: age, sex, body mass index (BMI), and body surface area (BSA) calculated with the Gehan and George formula; co-existing hypertension, diabetes mellitus, or dyslipidemia; smoking status; medical history of coronary artery disease; the European Heart Rhythm Association (EHRA) score of AF; co-existing dysthyroidism, obstructive pulmonary disease, or renal disease; medical history of stroke or transient ischemic attack; and pharmacological treatment. Coronary artery disease was diagnosed if patients had a history of myocardial infarction, percutaneous

coronary intervention, or coronary artery bypass grafting. Due to the subjectivity of the perception of arrhythmia, the duration of AF was not considered. In many cases, patients did not know when the arrhythmia started. The glomerular filtration rate (eGFR) was estimated using the Cockcroft-Gault formula. The CHA2DS2-VASc and HAS-BLED scores were registered according to the current European guidelines for AF treatment [2].

### Restoration of sinus rhythm

DCCV was performed under general sedation. Transesophageal echocardiography was performed to rule out the presence of thrombi in the left atrium. The DCCV was performed with paddles in an anterolateral position, using a biphasic defibrillator with the energy level at 150–300 J. If the first shock was ineffective, a second one was performed with an energy level that was higher by 100 J. The DCCV was considered successful if SR was achieved and maintained for at least 24 hours after the procedure. Patients in SR received anticoagulants, upstream therapy, or antiarrhythmic drugs according to their individual circumstances. A physician prescribed the antiarrhythmic drugs (amiodarone or propafenone) considering the risk of recurrence of AF but blinded to the echocardiographic parameters assessed in this study.

### Echocardiographic evaluation

Transthoracic echocardiography was performed according to current guidelines by an experienced echocardiographer using a Vivid S6 echocardiography machine (General Electric Medical Systems, Horten, Norway) with an M4S RS transducer [20, 21]. Standard M-mode Doppler imaging and two-dimensional cine loops of parasternal long- and short-axis, and apical two-, three-, and four-chamber views were obtained from each patient. All images and measurements were acquired from standard views and then stored. The digitally stored echocardiographic images were retrieved and analyzed with offline software (EchoPAC PC software, GE Medical Systems). The LA end-systolic volume (LAV) and end-diastolic volume (LAEDV) were measured from apical four- and two-chamber views using Simpson's method. The maximum volume of the left atrium (LAV) was measured on the frame just before mitral valve opening by tracing the inner border of the atrium, taking care to avoid the area under the valve annulus, the appendage, and the pulmonary veins. LAV was indexed to the BSA (LAVI). The minimum volume of the left atrium (LAEDV) was obtained on the frame of mitral valve closure and indexed to the BSA (LAEDVI). The LA emptying fraction (LAEF) was calculated with the following formula: (LA maximum volume–LA minimum volume)/LA maximum volume × 100%. The left ventricular (LV) volume and ejection fraction (LVEF) were assessed using Simpson's formula. The right atrial area (RAA) was assessed in the apical four-chamber view at the end of systole (RAAs) and at the end of diastole (RAAd) on the frame with tricuspid valve closure. A transmitral pulsed Doppler was recorded from an apical four-chamber view with a two-millimeter sample volume positioned between the tips of the mitral leaflets. A pulsed tissue Doppler imaging of the mitral annulus motion was performed from an apical 4-chamber view with a five-milliliter sample volume at the lateral and septal basal regions. The mean s' and e' were calculated as the averages of the septal and lateral measurements. The measurements obtained during AF were calculated by averaging the data from five consecutive beats.

### Statistical analysis

All of the statistical analyses of the echocardiographic parameters were made for measurements obtained during AF, before DCCV. The results are presented as mean ± standard deviation (SD) or as counts and percentages. Normally distributed variables were compared using

Student's t-test, and non-normally distributed variables were compared using the Mann-Whitney test or the chi-squared test. We ran univariate logistic regressions on the predictors of SR maintenance, and then analyzed the echocardiographic predictors that were statistically significant (p<0.1) with multivariate stepwise and forward logistic regressions. The stepwise inclusion was set at p<0.05 and exclusion at p>0.1. Moreover, we ran a multivariate logistic regression analysis with a model that included the independent echocardiographic predictors of SR maintenance from the previous analysis, clinical parameters with a p value <0.1 in the univariate logistic regression analysis, and important variables from a clinical point of view (age, hypertension, use of anti-arrhythmic drugs such as propafenone and amiodarone). Receiver operating characteristic (ROC) curves for predicting SR maintenance at one, six, and 12 months were calculated for selected echocardiographic variables. Optimal cut-offs were calculated based on Youden's J statistic, and areas under the curve (AUC) were compared using the DeLong test. Significance was set at p<0.05. All statistical analyses were performed with MedCalc Statistical Software version 18.6 (MedCalc Software Ltd, Ostend, Belgium).

## Results

There were 146 patients scheduled for elective DCCV due to persistent AF from July 2015 to August 2017. After DCCV, SR was restored in 117 (80.1%) patients. Of the 146 patients enrolled in the study, 61 (41.8%) patients maintained SR after 12 months of follow-up. The baseline characteristics of the study population are presented in Table 1. There were no differences in age, BMI, comorbidities, smoking habits, EHRA scale, CHA2DS2-VASc scale, HAS-BLED scale, or the use of antiarrhythmic drugs, statins, or the renin–angiotensin–aldosterone system blockade therapy before and after DCCV. Compared with patients with AF recurrence, patients who maintained SR at 12 months were more often male (72.1 vs. 54.1%; p = 0.028), had higher eGFR values (91±30 vs. 78.2±23.8 ml/min; p = 0.031), used beta-blockers more often before DCCV (98.4 vs. 84.7%; p = 0.006), and used diuretics less often before (31.1 vs. 55.3%; p = 0.004) and after DCCV (31.1 vs. 58.8%; p = 0.001). The echocardiographic parameters measured before DCCV are described in Table 2.

### Atrial enlargement

In the studied population, the mean LAVI was 47.8±12.4 ml/m$^2$, the mean LAEDVI was 36.6 ±12.3 ml/m$^2$, and the mean LAAP diameter was 44±4.5 mm. Patients with a smaller LA cavity were more likely to maintain SR. The group with SR maintenance and the group with AF recurrence had statistically significant differences in LAVI (44.3±11.5 vs. 50.3±12.5 ml/m$^2$; p = 0.004) and LAEDVI (30.5±11 vs. 37.6±12.5 ml/m$^2$; p<0.001), but there was no difference in LAAP diameter. The univariate logistic regression analysis revealed that elevated values of LAVI and LAEDVI decreased the odds of maintaining SR, with an odds ratio (OR) of 0.958 (95% CI 0.929–0.987; p = 0.006) and 0.946 (95% CI 0.916–0.978; p = 0.001), respectively. The mean RAA in the study population was 22.3±5.1 cm$^2$ in the systolic phase and 16.5±4.3 cm$^2$ in the diastolic phase. Among the parameters assessing RA enlargement, the RAAs and RAAd differed significantly between the group with maintained SR and the group with AF recurrence: 21.2±5.1 vs. 23.1±5 cm$^2$ (p = 0.015) and 15.8±4.4 vs. 17±4.1 cm$^2$ (p = 0.039), respectively. In the univariate logistic regression analysis, the OR of the RAAs for SR maintenance was 0.925 (95% CI 0.862–0.993; p = 0.03) and OR for the RAAd for SR maintenance was 0.932 (95% CI 0.859–1.012; p = 0.095).

### Emptying of the left atrium during atrial fibrillation

The mean emptying fraction during AF in the study population was 27.2±10% and was significantly different after 12 months between the group with SR maintenance and the group with

**Table 1. Clinical data of the study population at baseline and of the patients with sinus rhythm maintenance and atrial fibrillation recurrence after a 12-month follow-up.**

| | Study population n = 146 | SR maintenance n = 61 (41.8%) | Failure of DCCV or recurrence of AF n = 85 (58.2%) | p-value |
|---|---|---|---|---|
| Age (years) | 64.7±10.2 | 63±11.6 | 66±9 | 0.220 |
| Age <65 years (n, %) | 61 (41.8) | 29 (47.5) | 32 (37.6) | 0.234 |
| Age 65–74 years (n, %) | 64 (43.8) | 24 (39.3) | 40 (47.1) | 0.356 |
| Age ≥75 years (n, %) | 21 (14.4) | 9 (14.8) | 12 (14.1) | 0.914 |
| Males (n, %) | 90 (61.6) | 44 (72.1) | 46 (54.1) | 0.028 |
| BMI (kg/m$^2$) | 30.2±4.7 | 30.4±4.1 | 30.1±5.1 | 0.340 |
| Hypertension (n, %) | 122 (83.6) | 50 (82) | 72 (84.7) | 0.661 |
| Diabetes mellitus (n, %) | 29(19.9) | 13 (21.3) | 16 (18.8) | 0.711 |
| CAD stable (n, %) | 22 (15.1) | 10 (16.4) | 12 (14.1) | 0.701 |
| Heart failure (n, %) | 45 (30.8) | 21 (34.4) | 24 (28.2) | 0.426 |
| Stroke/TIA (n, %) | 14 (9.6) | 5 (8.2) | 9 (10.6) | 0.630 |
| Vascular disease (n, %) | 16 (11) | 8 (13.1) | 8 (9.4) | 0.481 |
| CHA2DS2-VASC | 2.7±1.6 | 2.6±1.5 | 2.8±1.6 | 0.318 |
| CHA2DS-VASC = 0 (n, %) | 9 (6.2) | 3 (4.9) | 6 (7.1) | 0.597 |
| CHA2DS2-VASC = 1 (n, %) | 27 (18.5) | 14 (23) | 13 (15.3) | 0.242 |
| CHA2DS-VASC ≥2 (n, %) | 110 (75.3) | 44 (72.1) | 66 (77.6) | 0.447 |
| HAS-BLED | 0.8±0.5 | 0.8±0.6 | 0.9±0.4 | 0.626 |
| Smokers (n, %) | 12 (8.2) | 4 (6.6) | 8 (9.4) | 0.537 |
| eGFR (ml/min) | 83.6±27.2 | 91±30 | 78.2±23.8 | 0.031 |
| Beta-blockers pre (n, %) | 132 (90.4) | 60 (98.4) | 72 (84.7) | 0.006 |
| Amiodarone pre (n, %) | 13 (8.9) | 4 (6.6) | 9 (10.6) | 0.401 |
| ACE inhibitors/ARB pre (n, %) | 119 (81.5) | 51 (83.6) | 68 (80) | 0.581 |
| Statins pre (n, %) | 95 (65.1) | 43 (70.5) | 52 (61.2) | 0.246 |
| Diuretics pre (n, %) | 66 (45.2) | 19 (31.1) | 47 (55.3) | 0.004 |
| Spironolactone/eplerenone pre (n, %) | 28 (19.2) | 16 (26.2) | 12 (14.1) | 0.068 |
| Beta-blockers post (n, %) | 118 (80.8) | 53 (86.9) | 65 (76.5) | 0.116 |
| Amiodarone post (n, %) | 48 (32.9) | 19 (31.1) | 29 (34.1) | 0.707 |
| Propafenone post (n, %) | 37 (25.3) | 20 (32.8) | 17 (20) | 0.080 |
| ACE/ARB post (n, %) | 122 (83.6) | 52 (85.2) | 70 (82.4) | 0.643 |
| Statins post (n, %) | 94 (64.4) | 40 (65.6) | 54 (63.5) | 0.800 |
| Diuretics post (n, %) | 69 (47.3) | 19 (31.1) | 50 (58.8) | 0.001 |
| Spironolactone/eplerenone post (n, %) | 31 (21.2) | 17 (27.9) | 14 (16.5) | 0.098 |

ACE inhibitors/ARB, angiotensin-converting-enzyme inhibitors/angiotensin II receptor blockers; AF, atrial fibrillation; BMI, body mass index; CAD, coronary artery disease; eGFR, glomerular filtration rate estimated from Cockcroft-Gault formula; HF, heart failure; pre, taken before cardioversion; post, taken after cardioversion; TIA, transient ischemic attack.

AF recurrence (30.8±8.3 vs. 24.6±10.4%; p<0.0001). In the univariate analysis, the OR of LAEF for SR maintenance was 1.072 (95% CI 1.031–1.115; p<0.001).

## Parameters of left ventricular filling pressure

During AF, the parameters that we can use to estimate LVFP are limited to the early filling wave (E), the early diastolic mitral annular velocity (e'), and the E/e' ratio. The mean values of these parameters in the study population are shown in Table 2. All of these parameters were significantly different in the group with SR maintenance and the group with AF recurrence. The early filling wave E was smaller in the group with SR maintenance compared to that

**Table 2. Echocardiographic parameters of the study population before cardioversion and of the patients with sinus rhythm maintenance and atrial fibrillation recurrence 12 months after cardioversion.**

| | Study population n = 146 | SR maintenance n = 61 (41.8%) | Failure of DCCV or recurrence of AF n = 85 (58.2%) | p-value |
|---|---|---|---|---|
| RVOTprox | 31±3.8 | 31.7±4.1 | 30.6±3.5 | 0.143 |
| IVS (mm) | 10.7±1.8 | 10.6±1.7 | 10.8±1.8 | 0.668 |
| LVEDD (mm) | 51.5±6.5 | 51.9±6.6 | 51.1±6.5 | 0.496 |
| LVESD (mm) | 36±7.6 | 36.8±8.2 | 35.5±7.2 | 0.313 |
| LVEDV (ml) | 118.4±35.1 | 125±35.7 | 113.6±34.1 | 0.053 |
| LVESV (ml) | 52.7±20.9 | 55.8±20 | 50.5±21.4 | 0.104 |
| LVSV (ml) | 77.6±23.2 | 67.5±20.5 | 63.5±21.3 | 0.166 |
| LVEF (%) | 56.8±10.4 | 55.3±9.6 | 57.8±10.9 | 0.223 |
| LAAP (mm) | 44±4.5 | 43.3±4.1 | 44.6±4.7 | 0.100 |
| LAVI (ml/m$^2$) | 47.8±12.4 | 44.3±11.5 | 50.3±12.5 | 0.004 |
| LAEDVI (ml/m$^2$) | 36.6±12.3 | 30.5±11 | 37.6±12.5 | <0.001 |
| LAEF (%) | 27.2±10 | 30.8±8.3 | 24.6±10.4 | <0.0001 |
| RAAs (cm$^2$) | 22.3±5.1 | 21.2±5.1 | 23.1±5 | 0.015 |
| RAAd (cm$^2$) | 16.5±4.3 | 15.8±4.4 | 17±4.1 | 0.039 |
| s'$_{mean}$ (cm/s) | 6±1.6 | 6.5±1.7 | 5.7±1.5 | 0.002 |
| e'$_{mean}$ (cm/s) | 10±2.3 | 10.6±2.3 | 9.5±2.3 | 0.011 |
| E/e'$_{mean}$ (cm/s) | 9.6±4 | 8.3±2.8 | 10.5±4.5 | 0.003 |
| E (m/s) | 0.9±0.2 | 0.8±0.2 | 0.9±0.2 | 0.004 |

E, early filling wave; e', early diastolic mitral annular velocity; LAAP, left atrial antero-posterior diameter; LAEF, left atrial emptying fraction; LAEDVI, left atrial end-diastolic volume index; LAVI, left atrial volume index; LVEDD, left ventricular end-diastolic diameter; LVESD, left ventricular end-systolic diameter; LVEDV, left ventricular end-diastolic volume; LVEF, left ventricular ejection fraction; LVESV, left ventricular end-systolic volume; LVSV, left ventricular stroke volume; RV prox, right ventricular proximal diameter; IVS, intraventricular septum wall thickness; RAA, right atrium area, d–diastolic, s–systolic.

measured in the AF recurrence group (0.8±0.2 vs. 0.9±0.2 m/s; p = 0.004). Conversely, the early diastolic mitral annular velocity (e'$_{mean}$) was higher in the group with SR maintenance than in the AF recurrence group (10.6±2.3 vs. 9.5±2.3 cm/s; p = 0.011). One of the most useful parameters to evaluate LVFP is the E/e'$_{mean}$ ratio, and it was also significantly smaller in the group of patients with SR maintenance than in the group with AF recurrence (8.3±2.8 vs. 10.5 ±4.5; p = 0.003). In the univariate logistic regression analysis, an increase in the E wave reduced the chance of SR maintenance, with an OR of 0.079 (95% CI 0.013–0.504; p = 0.007). An increase in the e'$_{mean}$ velocity of 1 cm/s increased the chance of SR maintenance by 22.8% (OR 1.228; 95% CI 1.049–1.438; p = 0.01). Greater values of the E/e'$_{mean}$ ratio decreased the odds of SR maintenance, with an OR of 0.84 (95% CI 0.753–0.938; p = 0.002).

## Left ventricular function

Because the study population included patients with normal function or mild systolic dysfunction of the left ventricle, we did not expect differences in the echocardiographic parameters of the LV function. Nevertheless, despite the non-statistically significant difference in the LVEF between the group with SR maintenance and the group with AF recurrence, the difference in the mitral annular peak systolic velocity (s') between the groups was significant (6.5±1.7 vs. 5.7 ±1.5 cm/s; p = 0.002).

## Multivariable logistic regression analysis and ROC curve analysis

In the univariate logistic regression, the following echocardiographic variables were significant predictors of SR maintenance at 12 months: LAVI, LAEDVI, LAEF, RAAs, RAAd, e'$_{mean}$, E/

**Table 3. Echocardiographic determinants of SR maintenance for 12 months according to the forward and stepwise multivariable regression analysis.**

| | Univariate analysis | | | Multivariable analysis | | |
|---|---|---|---|---|---|---|
| | OR | 95% CI | p-value | OR | 95% CI | p-value |
| LAVI (ml/m$^2$) | 0.958 | 0.929–0.987 | 0.006 | | | |
| LAEDVI (ml/m$^2$) | 0.946 | 0.916–0.978 | 0.001 | | | |
| LAEF (%) | 1.072 | 1.031–1.115 | <0.001 | 1.053 | 1.011–1.096 | 0.013 |
| RAAs (cm$^2$) | 0.925 | 0.862–0.993 | 0.030 | | | |
| RAAd (cm$^2$) | 0.932 | 0.859–1.012 | 0.095 | | | |
| e'$_{mean}$ (cm/s) | 1.228 | 1.049–1.438 | 0.010 | | | |
| E/e'$_{mean}$ | 0.840 | 0.753–0.938 | 0.002 | 0.883 | 0.788–0.990 | 0.033 |
| E (m/s) | 0.079 | 0.013–0.504 | 0.007 | | | |

E, early filling wave; e', early diastolic mitral annular velocity; LAEF, left atrial emptying fraction; LAEDVI, left atrial end-diastolic volume index; LAVI, left atrial volume index; RAA, right atrium area; d, diastolic, s, systolic.

e'$_{mean}$ ratio, and E wave. In the multivariate forward and stepwise logistic regression analysis, only LAEF (OR 1.053; 95% CI 1.011–1.096; p = 0.013) and the E/e'$_{mean}$ ratio (OR 0.883; 95% CI 0.788–0.99; p = 0.033) remained significant predictors of SR maintenance (Table 3). In the multivariate logistic regression analysis including echocardiographic and clinical variables, only LAEF (OR 1.046; 95% CI 1–1.095; p = 0.049) and beta-blocker use before DCCV (OR 14.694; 95% CI 1.622–133.139; p = 0.017) were significant predictors of SR maintenance at 12 months (Table 4). In the ROC curve analysis, LAEF before DCCV had an AUC for the prediction of SR maintenance after 12 months of 0.680 (p<0.001), with a cut-off value of 23.9%, sensitivity of 83.6%, and specificity of 51.2% (Fig 1). The ROC curve analysis for LAEF following successful DCCV at the first, sixth, and 12$^{th}$ month revealed an AUC that increased over the time of observation. The ROC curve analysis for the E/e'$_{mean}$ ratio revealed an AUC of 0.645 (p = 0.002), with a cut-off value of 8.7, sensitivity of 73.8%, and specificity of 55.4% (Fig 2). The AUC increased over the time of observation.

## Discussion

Our results demonstrate that when LAEF and parameters assessing LVFP are measured during AF before DCCV, they are valuable predictors of SR maintenance after DCCV. Additionally,

**Table 4. Echocardiographic and clinical determinants of SR maintenance for 12 months according to the multivariable regression analysis.**

| | Univariate analysis | | | Multivariable analysis | | |
|---|---|---|---|---|---|---|
| | OR | 95% CI | p-value | OR | 95% CI | p-value |
| LAEF (%) | 1.072 | 1.031–1.115 | <0.001 | 1.046 | 1–1.095 | 0.049 |
| E/e'$_{mean}$ | 0.840 | 0.753–0.938 | 0.002 | | | |
| Age (years) | 0.970 | 0.939–1.003 | 0.078 | | | |
| Males | 2.194 | 1.086–4.436 | 0.029 | | | |
| eGFR (ml/min) | 1.018 | 1.005–1.032 | 0.008 | | | |
| Hypertension | 0.821 | 0.340–1.979 | 0.660 | | | |
| Beta-blockers pre | 9.833 | 1.242–77.839 | 0.030 | 14.694 | 1.622–133.139 | 0.017 |
| Diuretics pre | 0.361 | 0.179–0.730 | 0.005 | | | |
| Amiodaron post | 0.955 | 0.438–2.083 | 0.908 | | | |
| Propafenone post | 1.119 | 0.512–2.444 | 0.778 | | | |

E, early filling wave; e', early diastolic mitral annular velocity; LAEF, left atrial emptying fraction; eGFR, estimated glomerular filtration rate; post, taken after cardioversion; pre, taken before cardioversion.

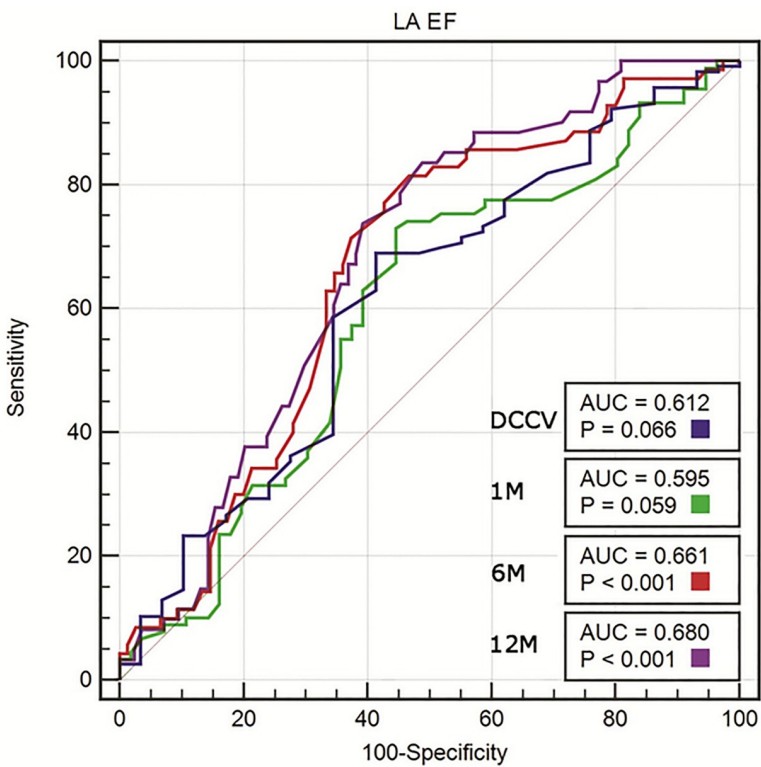

**Fig 1. ROC curve analysis of LAEF measured before cardioversion for predicting the success of electrical cardioversion (DCCV) and SR maintenance at 1, 6, and 12 months.** AUC, area under the curve; p-vales for AUC comparisons with no effect (AUC = 0.5).

the prognostic value of LAEF in the context of SR maintenance increased along with observation time. To the best of our knowledge, this is the first study showing that LAEF measured during AF allows for the assessment of the prognosis of SR maintenance after DCCV.

To date, LA contractility during AF has been considered to be so disturbed as to not affect LAEF, independently of the state of LA remodeling. Our results indicate that LA emptying volume is still generated despite a disturbed contractility of the left atrium during AF, and this emptying volume has a prognostic value in SR maintenance after DCCV. LA myocardial contractility is influenced by the mechanical remodeling of the left atrium, and patients who maintained SR had a higher LAEF. Therefore, we can conclude that the mechanical remodeling of the left atrium was less advanced in these patients. Furthermore, we showed that the echocardiographic parameters assessing mechanical remodeling (LAEF) have a greater prognostic value for SR maintenance after DCCV than parameters assessing structural remodeling (LAVI, LAEDVI).

De Vos et al. showed that LA myocardial contractility, assessed as the velocity of the LA wall measured during AF, is associated with the short- and long-term prognosis of SR maintenance after DCCV [22]. They demonstrated that mechanical remodeling, assessed by measuring LA myocardial velocity, influences the efficiency of DCCV. They also described the relationship between the duration of AF and LA myocardial velocity and thus demonstrated the effect of AF duration on the mechanical remodeling of the left atrium [23]. Based on these studies, we hypothesized that patients with less advanced mechanical remodeling have higher LAEF during AF, and thus have a better prognosis for SR maintenance after DCCV. Patients with less advanced mechanical remodeling of the left atrium have better contractility of LA

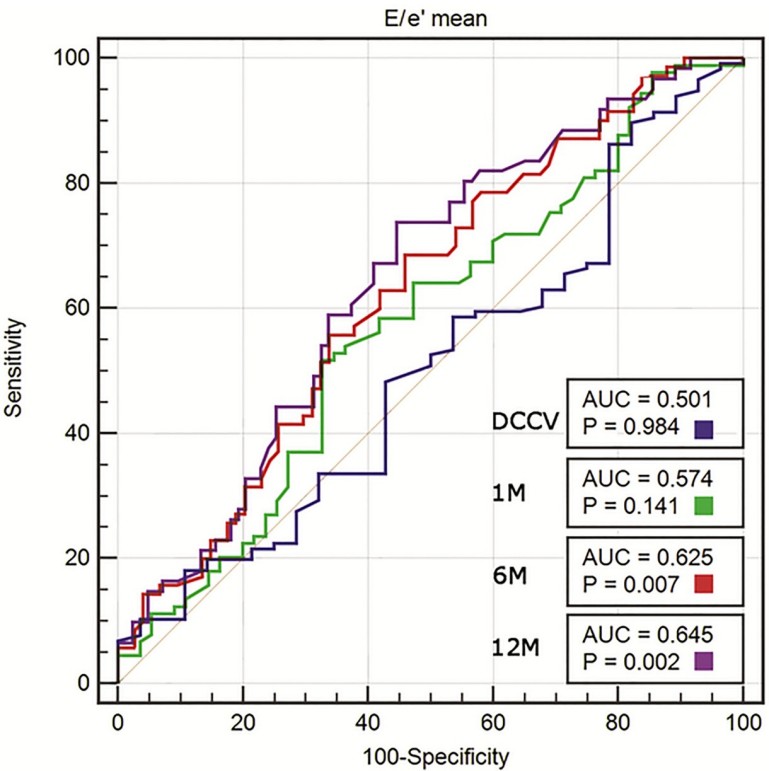

**Fig 2. ROC curve analysis of E/e' mean measured before cardioversion for predicting the success of electrical cardioversion (DCCV) and SR maintenance at 1, 6, and 12 months.** AUC, area under the curve; p-vales for AUC comparisons with no effect (AUC = 0.5).

myocardium compared with those with more advanced remodeling. This results in a greater volume of blood transported to the left ventricle during ventricular diastole and a smaller volume of blood remaining in the left atrium. Kim et al. tested a similar hypothesis, but they focused on the blood flow wave through the mitral valve directly after the E wave (early diastolic mitral inflow): the left atrial fibrillatory contraction flow (Afc) [24]. They concluded that the presence and nature of the Afc wave indicate advanced mechanical remodeling of the left atrium and thus allow for the assessment of the risk of AF recurrence after DCCV. The higher the velocity of Afc and the Afc integral, the lower the risk of AF recurrence. These studies support our hypothesis about the relationship between mechanical remodeling, LA wall contractility assessed during AF, and LAEF assessed during AF.

The most common risk factors for AF recurrence after DCCV are reflected in structural remodeling parameters like LAAP, LAEDVI, or LAVI [13, 14]. However, an increasing number of studies indicate that mechanical remodeling parameters measured by conventional or new techniques are more accurate to evaluate the risk of AF recurrence after DCCV [16, 25–27]. Luong et al. showed that RAEF and LAEF are better predictors of AF recurrence after DCCV than RAVI or LAVI. LA contractility during SR is also manifested by E wave velocity. Spiecker et al. and Grundvold et al. showed that A wave peak velocity is a risk factor for AF recurrence after DCCV [28, 29]. Also, LA strain measured with tissue Doppler echocardiography [25, 30] and speckle tracking echocardiography (STE) [31, 32] and myocardial velocity of the left atrium appendage measured during AF [27] have prognostic value for SR maintenance after DCCV. Currently, the STE technique is preferred for the evaluation of the cardiac strain and strain rate because it is free from an error due to the angular relationship of the

measurements. The prognostic value of LA strain regarding prognosis of SR maintenance in patients with AF is still being studied. Some reports showed the prognostic value of LA strain in terms of SR maintenance after DCCV [30–32], while some showed only the prognostic value of the dispersion of time to the maximal longitudinal strain of LA segments but not the maximum strain value [26]. LA strain measurements are also used to assess the prognosis of SR maintenance after ablation of pulmonary vein isolation due to AF [33]. Also new techniques such as three-dimensional echocardiography were used to assess the LA phasic function and has been shown that the conduit function assessed using this technique has a prognostic value in terms of AF recurrence after DCCV [34].

Multivariable analysis showed that the E/e'$_{mean}$ ratio measured during AF is the second independent risk factor for AF recurrence. This parameter reflects LVFP and can be measured during SR and AF. Similarly to our study, most published studies have focused on LVFP measurement during AF before DCCV [17–19]. The assessment of LVFP during SR following a successful DCCV has prognostic value for SR maintenance after DCCV [35].

LAEF measured during AF is simple to calculate but requires the performance of measurements in several subsequent heart cycles, which can complicate its introduction to common clinical practice. The relationship between LAEF measured during AF and the efficiency of DCCV further confirms the superiority of mechanical remodeling parameters over structural remodeling parameters in the risk assessment of AF recurrence after DCCV. Measuring mechanical remodeling parameters can help with the qualification of patients to SR maintenance strategies using antiarrhythmic drugs, DCCV, or AF ablation.

## Study limitations

Our study was carried out in only one center and with a small sample, although it is one of the largest studies to date on echocardiographic predictors of SR maintenance after DCCV due to AF. When interpreting our results, one should remember that echocardiography is operator-dependent and requires experience and skill. Therefore, all echocardiographic measurements in this study were made by one experienced investigator. We did not measure RAEF, which could be a predictor of SR maintenance after DCCV due to AF. In addition, we assessed AF duration retrospectively based on the patients' reports. Because this method is unreliable, we did not analyze AF duration as a predictor of SR maintenance after successful DCCV. Moreover, because constant heart rhythm monitoring was not feasible in our long-term study, we could have missed self-limiting episodes of AF recurrence. We also performed all DCCVs in the antero-lateral position, without changing the paddle position if DCCV failed, which might have influenced the success rate.

## Conclusions

LAEF, measured during AF, is an independent prognostic parameter for SR maintenance in the first 12 months following DCCV due to persistent AF. We are the first to present evidence of its predictive value in this context. LAEF and the E/e'$_{mean}$ ratio were independent parameters allowing for the prediction of SR maintenance after DCCV. Our results support the use of LA mechanical remodeling parameters over structural remodeling parameters to assess the prognosis of SR maintenance after DCCV. The assessment of LA remodeling can help qualify patients to SR maintenance strategies or ventricular rate control strategies.

## Supporting information

**S1 Data.**
(XLSX)

## Author Contributions

**Conceptualization:** Paweł Wałek, Beata Wożakowska-Kapłon.

**Data curation:** Paweł Wałek, Iwona Gorczyca, Janusz Sielski.

**Formal analysis:** Paweł Wałek, Beata Wożakowska-Kapłon.

**Funding acquisition:** Paweł Wałek, Beata Wożakowska-Kapłon.

**Investigation:** Paweł Wałek, Janusz Sielski, Beata Wożakowska-Kapłon.

**Methodology:** Paweł Wałek, Beata Wożakowska-Kapłon.

**Project administration:** Paweł Wałek, Janusz Sielski.

**Resources:** Paweł Wałek, Iwona Gorczyca.

**Software:** Paweł Wałek.

**Supervision:** Beata Wożakowska-Kapłon.

**Validation:** Paweł Wałek.

**Visualization:** Paweł Wałek.

**Writing – original draft:** Paweł Wałek, Beata Wożakowska-Kapłon.

**Writing – review & editing:** Paweł Wałek, Iwona Gorczyca, Beata Wożakowska-Kapłon.

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
