## [Decision Letter · Decision Letter 0]

3 Jun 2020

PONE-D-20-14322

Left atrial emptying fraction determined during atrial fibrillation predicts maintenance of sinus rhythm after direct current cardioversion in patients with persistent atrial fibrillation

PLOS ONE

Dear Dr. Gorczyca,

Thank you for submitting your manuscript to PLOS ONE. After careful consideration, we feel that it has merit but does not fully meet PLOS ONE’s publication criteria as it currently stands. Therefore, we invite you to submit a revised version of the manuscript that addresses the points raised during the review process.

Please address all the issues raised by the reviewers.

We look forward to receiving your revised manuscript.

Kind regards,

Elena Cavarretta, M.D., Ph.D.

Academic Editor

PLOS ONE

Journal Requirements:

Reviewers' comments:

Reviewer's Responses to Questions

**Comments to the Author**

1. Is the manuscript technically sound, and do the data support the conclusions?

Reviewer #1: Yes

Reviewer #2: Yes

2. Has the statistical analysis been performed appropriately and rigorously? 

Reviewer #1: Yes

Reviewer #2: Yes

3. Have the authors made all data underlying the findings in their manuscript fully available?

Reviewer #1: Yes

Reviewer #2: Yes

4. Is the manuscript presented in an intelligible fashion and written in standard English?

Reviewer #1: Yes

Reviewer #2: Yes

5. Review Comments to the Author

Reviewer #1: Thank you for inviting me to review this article. I read the manuscript by Gorczyca et al. with interest.

I appreciated the general purpose of this paper, firstly because the research of reliable indices predicting atrial fibrillation (AF) recurrence after electric cardioversion (ECV) is important to guide therapeutic strategies, especially in some case of older and asymptomatic patients when rate control therapy could be preferred; secondly, because I totally agree with the superiority of bi-dimensional measures of left atrial (LA) volume than mono-dimensional LA diameters for the evaluation of LA structure in order to obtain relevant diagnostic and prognostic information. LA emptying fraction (LAEF) could represent an additive useful parameter to consider a dynamic echocardiographic index of LA mechanics.

However, the small sample size and the low-medium statistical strength of the results (AUC 0.68, with very low specificity-51.2%) which is, in my opinion, due to the absence of newest advanced and more accurate methods (e.g. speckle tracking or 3D echocardiography,), limits the importance of this findings. I think this should be taken cautiously and used only if more advance techniques are not available.

The paper is overall well written. The statistical methods are appropriate and clearly explained.

As major comments:

• In the introduction, you wrote that ECV “can be a bridging therapy until AF ablation is performed [2]”. This sentence should be rephrased. In fact, it seems that you mean that AF ablation is the only therapeutic option and sometimes ECV could be used as bridging therapy. This is not correct, since the choice of AF ablation therapy should be tailored on patients’ characteristics and symptoms.

• An additive sub-analysis investigating the rate of recurrence depending on the antiarrhythmic drug administered after ECV would be useful.

• Could you precisely list the parameters considered in multivariate analysis? You only wrote “echocardiographic parameters” in the methods section.

• Since we know that “diuretic therapy” not only could influence LA volumes (and consequently LAEF), but also it showed a statistically significant variation between groups and value at univariate analysis, I think it is important to include it into multivariate analysis. If you have already done it, please highlight it into the results/discussion section, if not, please do it as a further analysis.

• Did you assess sinus rhythm maintenance only with a spot-EKG or with a 24-hours Holter EKG registration? Please precisely describe it in the methods section.

• Advanced imaging methods, such as speckle tracking or 3D echocardiography, showed higher AUC for the prediction of sinus rhythm maintenance after ECV, and to provide prognostic information. Nowadays, these are also easily available and low time-consuming than other advanced imaging methods. Why didn’t you perform them? Please at least discuss their potential role in the discussion section.

Adding references to these articles could be useful:

PMID: 30993507

PMID: 32219615

PMID: 30829875

As minor comments:

• How do you explain the significance of eGFR in multivariate analysis? Could you discuss it?

• Page 13, line 2 and 24: please use the abbreviation LA for “left atrial”

• Page 14 line 4 , please add a space between “and” and “LAEF”

Reviewer #2: COMMENTS FOR THE AUTHORS

Walek et al report a prospective cohort study to identify predictors of sinus rhythm maintenance after direct current cardioversion.

The paper is well-structured and most of the statements are easy to follow.

The results show left atrial emptying fraction, measured during atrial fibrillation, as an independent prognostic parameter for sinus rhythm maintenance in the first 12 months following cardioversion and highlight the role of left atrial mechanical remodeling parameters over structural ones.

However, some major and minor sources of weakness in the reliability of these results must be considered.

Major corrections:

- Patients with high ventricular response to atrial fibrillation were counted in the exclusion criteria? Under this circumstance, the assessment of left atrial function is challenging. A cut-off of mean ventricular rate was taken into account?

- In the Results section, the number of patients with sinus rhythm maintenance/recurrence of atrial fibrillation is not clear with incongruity between the text and the Tables 1 and 2.

- In the paragraph “Atrial enlargement” LAESVI is mentioned many times, instead of left atrial end-diastolic volume index (LAEDVI).

- In the same paragraph, there are some mistakes in the statistical connections of the sentence “The univariate logistic regression analysis revealed that elevated values of LAVI and LAEDVI decreased the odds of maintaining SR, with an odds ratio (OR) of 0.958 (95%CI 0.929-0.987; p=0.006) for the patients that maintained SR and 0.946 (95%CI 0.916-0.978; p=0.001) for those with AF recurrence”.

Minor corrections:

- In the Abstract, the sentence “Analysis with a model including clinical and echocardiographic variables, only LAEF …” is clearer writing “Analyzing a model including clinical and echocardiographic variables, only LAEF …”.

- Make sure to report the same numerical data in the text and in Tables; there are a few discrepancies between the data typed in the text and in the Tables (for example: page 9: 37.6±12.5 ml/m2 in the text vs 37.6±12.3 ml/m2 in Table 2; page 10: 95%CI 0.735-0.938 in the text vs 95%CI 0.753-0.938 in Table 3).

- At page 9 there is a typo: 95%CI 0.8859-1.012, correct form: 0.859.

- At page 9 the sentence “In the univariate analysis, the OR for LAEF for the SR maintenance group was…” is clearer writing “In the univariate analysis, the OR of LAEF for SR maintenance was …”.

- In the Figure legends at page 12 probably there is a typo: “p values”.

- In the last sentence of the Conclusions, the correct form is “ventricular rate control”.

6. PLOS authors have the option to publish the peer review history of their article (what does this mean?). If published, this will include your full peer review and any attached files.

Reviewer #1: No

Reviewer #2: No

---

## [Author Response · Author response to Decision Letter 0]

30 Jul 2020

Dear Reviewers,

Thank you very much for a thorough review of our manuscript and valuable comments. We would also like to apologize for minor editorial errors. We hope that our amendments will meet your acceptance. 

We responded to the comments as follows:

Reviewer I

1. In the introduction, you wrote that ECV “can be a bridging therapy until AF ablation is performed [2]”. This sentence should be rephrased. In fact, it seems that you mean that AF ablation is the only therapeutic option and sometimes ECV could be used as bridging therapy. This is not correct, since the choice of AF ablation therapy should be tailored on patients’ characteristics and symptoms.

In introduction the sentence has been changed 

“and could be a bridging therapy until AF ablation is performed in a selected group of patients”

2. An additive sub-analysis investigating the rate of recurrence depending on the antiarrhythmic drug administered after ECV would be useful. 

Due to the insignificant value of univariate regression analysis of anti-arrhythmic drugs such as propafenone or amiodarone, we did not perform additional analyzes for these variables. Despite the insignificant value in the univariate analysis, taking into account the clinical significance of antiarrhythmic drugs in the pharmacotherapy of patients with AF, we included these variables in the multivariate analysis model (Table 4) but also in this analysis these variables did not reach a statistically significant value. Of the anti-arrhythmic drugs, only beta-blockers were independent predictors of SR maintenance after 12 months.

In methods the sentence has been added:

 “…and important variables from a clinical point of view (age, hypertension, use of anti-arrhythmic drugs such as propafenone and amiodarone)”

3. Could you precisely list the parameters considered in multivariate analysis? You only wrote “echocardiographic parameters” in the methods section.

All echocardiographic parameters that we have included in the multivariable analysis are listed in Table 3. These parameters are listed in the multivariable logistic regression analysis and ROC curve analysis chapter. Listing them in the method chapter will generate replicates. At the explicit request of the reviewer, we can list the parameters from Table 3 in brackets in the statistical analysis chapter. 

 4. Since we know that “diuretic therapy” not only could influence LA volumes (and consequently LAEF), but also it showed a statistically significant variation between groups and value at univariate analysis, I think it is important to include it into multivariate analysis. If you have already done it, please highlight it into the results/discussion section, if not, please do it as a further analysis.

The use of diuretics before cardioversion was included in the multivariable analysis (Table 4 - "diuretics pre"). This is also described in the results chapter.

“Compared with patients with AF recurrence, patients who maintained SR at 12 months were more often male (72.1 vs. 54.1%; p=0.028), had higher eGFR values (91±30 vs. 78.2±23.8 ml/min; p=0.031), used beta-blockers more often before DCCV (98.4 vs. 84.7%; p=0.006), and used diuretics less often before (31.1 vs. 55.3%; p=0.004) and after DCCV (31.1 vs. 58.8%; p=0.001)”.

5. Did you assess sinus rhythm maintenance only with a spot-EKG or with a 24-hours Holter EKG registration? Please precisely describe it in the methods section. 

The follow-up has been described in study population chapter.

“Follow-up electrocardiograms and clinical data were collected from all patients with SR after one month, six months, and 12 months. A 24-hour ambulatory electrocardiographic monitoring was performed on all patients who were in SR at the one- and 12-month follow-ups”.

6. Advanced imaging methods, such as speckle tracking or 3D echocardiography, showed higher AUC for the prediction of sinus rhythm maintenance after ECV, and to provide prognostic information. Nowadays, these are also easily available and low time-consuming than other advanced imaging methods. Why didn’t you perform them? Please at least discuss their potential role in the discussion section. Adding references to these articles could be useful:

PMID: 30993507 

PMID: 32219615 

PMID: 30829875 

Thank you very much for comments on the discussion, especially for the article on 3D echocardiography. The discussion has been expanded to include the following sentences.

“Also, LA strain measured with tissue Doppler echocardiography [25, 30] and speckle tracking echocardiography (STE) [31, 32] and myocardial velocity of the left atrium appendage measured during AF [27] have prognostic value for SR maintenance after DCCV. Currently, the STE technique is preferred for the evaluation of the cardiac strain and strain rate because it is free from an error due to the angular relationship of the measurements. The prognostic value of LA strain regarding prognosis of SR maintenance in patients with AF is still being studied. Some reports showed the prognostic value of LA strain in terms of SR maintenance after DCCV [30-32], while some showed only the prognostic value of the dispersion of time to the maximal longitudinal strain of LA segments but not the maximum strain value [26]. LA strain measurements are also used to assess the prognosis of SR maintenance after ablation of pulmonary vein isolation due to AF [33]. Also new techniques such as three-dimensional echocardiography were used to assess the LA phasic function and has been shown that the conduit function assessed using this technique has a prognostic value in terms of AF recurrence after DCCV [34]”.

Moreno-Ruiz's article was already in the earlier version of the manuscript under number 32.

References have been added:

33. Pastore MC, De Carli G, Mandoli GE, D'Ascenzi F, Focardi M, Contorni F et al .The prognostic role of speckle tracking echocardiography in clinical practice: evidence and reference values from the literature. Heart Fail Rev. 2020 Mar 26. doi: 10.1007/s10741-020-09945-9. [Epub ahead of print].

34. Giubertoni A, Boggio E, Ubertini E, Zanaboni J, Calcaterra E, Degiovanni A et al. Atrial conduit function quantitation precardioversion predicts early arrhythmia recurrence in persistent atrial fibrillation patients. J Cardiovasc Med (Hagerstown). 2019;20:169-179.

7. How do you explain the significance of eGFR in multivariate analysis? Could you discuss it? 

eGFR was statistically significant only in univariate analysis. eGFR was not statistically significant in multivariable analysis in a model containing clinical and echocardiographic parameters (Table 4).

8. Page 13, line 2 and 24: please use the abbreviation LA for “left atrial”.

The sentence has been corrected.

9. Page 14 line 4 , please add a space between “and” and “LAEF”.

The sentence has been corrected.

Reviewer II

1. Patients with high ventricular response to atrial fibrillation were counted in the exclusion criteria? Under this circumstance, the assessment of left atrial function is challenging. A cut-off of mean ventricular rate was taken into account?

Thank you for your comment. Due to the fact that all patients included in the study were patients admitted to elective DCCV, they had optimally controlled ventricular rate. Due to the significant value of information about the ventricular rate, we added "ventricular rate greater than 120 beats per minute" to the exclusion criteria because among the patients in our study the upper limit of the average ventricular rate during AF before DCCV was 120 beats per minute.

In the study population chapter, the sentence has been changed.

“Exclusion criteria were as follows: age under 18 years, lack of consent for study participation, lack of consent for DCCV, poor quality of echocardiography visualization, ventricular rate greater than 120 beats per minute, moderate or severe valve regurgitation or stenosis…”. 

2. In the Results section, the number of patients with sinus rhythm maintenance/recurrence of atrial fibrillation is not clear with incongruity between the text and the Tables 1 and 2.

The sentences have been corrected. We apologize for this mistake.

“Of the 146 patients enrolled in the study, 61 (41,8%) patients maintained SR after 12 months of follow-up”.

3. In the paragraph “Atrial enlargement” LAESVI is mentioned many times, instead of left atrial end-diastolic volume index (LAEDVI).

The sentences have been corrected. We apologize for the mistake.

“In the studied population, the mean LAVI was 47.8±12.4 ml/m², the mean LAEDVI was 36.6±12.3 ml/m², and the mean LAAP diameter was 44±4.5 mm. Patients with a smaller LA cavity were more likely to maintain SR. The group with SR maintenance and the group with AF recurrence had statistically significant differences in LAVI (44.3±11.5 vs. 50.3±12.5ml/m²; p=0.004) and LAEDVI (30.5±11 vs. 37.6±12.5 ml/m²; p<0.001), but there was no difference in LAAP diameter”.

4. In the same paragraph, there are some mistakes in the statistical connections of the sentence “The univariate logistic regression analysis revealed that elevated values of LAVI and LAEDVI decreased the odds of maintaining SR, with an odds ratio (OR) of 0.958 (95%CI 0.929-0.987; p=0.006) for the patients that maintained SR and 0.946 (95%CI 0.916-0.978; p=0.001) for those with AF recurrence”.

The sentences have been corrected. We apologize for the mistake.

“The univariate logistic regression analysis revealed that elevated values of LAVI and LAEDVI decreased the odds of maintaining SR, with an odds ratio (OR) of 0.958 (95%CI 0.929-0.987; p=0.006) and 0.946 (95%CI 0.916-0.978; p=0.001) respectively”. 

5. In the Abstract, the sentence “Analysis with a model including clinical and echocardiographic variables, only LAEF …” is clearer writing “Analyzing a model including clinical and echocardiographic variables, only LAEF …”.

The sentences have been corrected.

6. Make sure to report the same numerical data in the text and in Tables; there are a few discrepancies between the data typed in the text and in the Tables (for example: page 9: 37.6±12.5 ml/m2 in the text vs 37.6±12.3 ml/m2 in Table 2; page 10: 95%CI 0.735-0.938 in the text vs 95%CI 0.753-0.938 in Table 3).

The sentences have been corrected.

7. At page 9 there is a typo: 95%CI 0.8859-1.012, correct form: 0.859.

The sentence has been corrected.

8. At page 9 the sentence “In the univariate analysis, the OR for LAEF for the SR maintenance group was…” is clearer writing “In the univariate analysis, the OR of LAEF for SR maintenance was …”.

The sentence has been corrected.

9. In the Figure legends at page 12 probably there is a typo: “p values”.

The sentences have been corrected.

10. In the last sentence of the Conclusions, the correct form is “ventricular rate control”.

The sentence has been corrected.

Yours faithfully,

Authors

---

## [Decision Letter · Decision Letter 1]

7 Aug 2020

Left atrial emptying fraction determined during atrial fibrillation predicts maintenance of sinus rhythm after direct current cardioversion in patients with persistent atrial fibrillation

PONE-D-20-14322R1

Dear Dr. Wałek,

We’re pleased to inform you that your manuscript has been judged scientifically suitable for publication and will be formally accepted for publication once it meets all outstanding technical requirements.

Kind regards,

Elena Cavarretta, M.D., Ph.D.

Academic Editor

PLOS ONE

Additional Editor Comments (optional):

Reviewers' comments:

Reviewer's Responses to Questions

**Comments to the Author**

1. If the authors have adequately addressed your comments raised in a previous round of review and you feel that this manuscript is now acceptable for publication, you may indicate that here to bypass the “Comments to the Author” section, enter your conflict of interest statement in the “Confidential to Editor” section, and submit your "Accept" recommendation.

Reviewer #1: All comments have been addressed

Reviewer #2: All comments have been addressed

2. Is the manuscript technically sound, and do the data support the conclusions?

Reviewer #1: Yes

Reviewer #2: Yes

3. Has the statistical analysis been performed appropriately and rigorously? 

Reviewer #1: Yes

Reviewer #2: Yes

4. Have the authors made all data underlying the findings in their manuscript fully available?

Reviewer #1: (No Response)

Reviewer #2: Yes

5. Is the manuscript presented in an intelligible fashion and written in standard English?

Reviewer #1: Yes

Reviewer #2: Yes

6. Review Comments to the Author

Reviewer #1: (No Response)

Reviewer #2: The Authors have fulfilled nearly all the requests in the revised version of the manuscript. Only the typo “p values” in the Figure legends at page 12 is pending.

7. PLOS authors have the option to publish the peer review history of their article (what does this mean?). If published, this will include your full peer review and any attached files.

Reviewer #1: No

Reviewer #2: No

---

## [Editor Report · Acceptance letter]

12 Aug 2020

PONE-D-20-14322R1 

Left atrial emptying fraction determined during atrial fibrillation predicts maintenance of sinus rhythm after direct current cardioversion in patients with persistent atrial fibrillation 

Dear Dr. Wałek:

I'm pleased to inform you that your manuscript has been deemed suitable for publication in PLOS ONE. Congratulations! Your manuscript is now with our production department. 

Kind regards, 

on behalf of

Dr. Elena Cavarretta 

Academic Editor

PLOS ONE